# High Heterogeneity of Temporal Bone CT Aspects in Osteogenesis Imperfecta Is Not Linked to Hearing Loss

**DOI:** 10.3390/jcm11082171

**Published:** 2022-04-13

**Authors:** Aïcha Ltaief-Boudrigua, Genevieve Lina-Granade, Eric Truy, Ruben Hermann, Guillaume Chevrel

**Affiliations:** 1Department of Radiology, Edouard Herriot Hospital, Hospices Civils de Lyon, 69003 Lyon, France; aicha.ltaief-boudrigua@chu-lyon.fr; 2Department of Oto-Rhino-Laryngology, Head and Neck Surgery, Edouard Herriot Hospital, Hospices Civils de Lyon, 69003 Lyon, France; genevieve.lina-granade@chu-lyon.fr (G.L.-G.); eric.truy@chu-lyon.fr (E.T.); ruben.hermann@chu-lyon.fr (R.H.); 3Centre de Compétence Maladies Rares en ORL, Hospices Civils de Lyon, 69003 Lyon, France; 4INSERM U1028, CNRS UMR5292, Lyon Neuroscience Research Center, Equipe IMPACT, 69675 Bron, France; 5Claude Bernard University Lyon 1, 69008 Lyon, France; 6Department of SAMU 69-Emergency, Edouard Herriot Hospital, Hospices Civils de Lyon, 69003 Lyon, France; 7Department of Rheumatology and Bone Diseases, Edouard Herriot Hospital, Hospices Civils de Lyon, 69003 Lyon, France

**Keywords:** osteogenesis imperfecta, hearing loss, temporal bone, mutation, temporal bone computed tomography, bone mineral density

## Abstract

Objectives: To determine whether temporal bone computed tomography (CT) features are linked to the presence and type of hearing loss in osteogenesis imperfecta (OI) when considering hearing-impaired OI patients and normally hearing (NH) OI ones. A secondary objective was to assess whether other factors influence CT features in a large sample: age, type of mutation, or bone mineral density (BMD). Methods: A total of 41 adults with OI underwent CTs and pure-tone audiometry in 82 ears. Hearing thresholds were normal in 64 out of 82 ears, and most had not been operated on for stapedectomy or stapedotomy. Ossicle density, footplates, oval and round windows, retrofenestral peri- and endolabyrinths, and temporal pneumatization were analyzed twice by an experienced radiologist. CT features were compared to hearing, age, collagen mutations, and bone mineral density. Results: Unexpectedly a high prevalence of footplate, ossicle, and otic capsule anomalies was observed, even in NH ears. Footplate hypodensity or thickening was mostly found in ears without conductive hearing loss. There were significantly more retrofenestral anomalies or window obstruction in ears with a sensorineural hearing loss component than in ears without. Age was significantly higher in ears with middle layer hypodensity than in ears without. Patients with mutations were expected to have reduced collagen quantity and had significantly more footplate or retrofenestral anomalies than those with qualitative mutations. BMD was significantly higher in ears without temporal hyperpneumatization. Conclusion: Temporal bone CT features in OI are present in a large proportion of patients, had they hearing loss or not, and might be determined more by collagen mutation type than by age or BMD.

## 1. Introduction

Osteogenesis imperfecta (OI) is a type-1 collagen genetic disease that results in abnormal bone fragility [1,2]. It is mostly, but not exclusively, caused by an autosomal-dominant mutation in the COL1A1 or COL1A2 genes. The main manifestations are bone fractures and deformities, short stature, and blue sclerae. Moreover, many OI patients suffer from progressive hearing loss, reported in 26% to 60% of patients, depending on age and OI type [3,4,5].

The pathology, natural history, and treatment of hearing loss in OI resembles that of otosclerosis (OS). Computed tomography (CT) reveals fenestral, antefenestram, and otic capsule hypodensities [6,7,8]. Hearing loss is usually conductive at the beginning, which is related to stapes footplate (FP) ankylosis, and could be surgically cured by either stapedotomy or stapedectomy [9]. Secondary or isolated sensorineural hearing damage occurs in a larger proportion of OI patients than in OS ones. By analogy with OS, extensive retrofenestral CT lesions (in an otic capsule) are considered to be responsible for SNCs and might lead to the rejection of surgical treatment, with hearing aids being the only possible rehabilitation [6,7,8].

However, correlation between CT findings and hearing loss is uncertain: Swinnen et al., firstly, comparing hearing and CT scans, observed discrepancies between CT and hearing in 27.8% (17 patients) [10]. In this series, most patients had hearing loss. No studies to our knowledge describe temporal bone (TB) CT in normally hearing (NH) OI patients.

The aims of this study were: (i) to determine whether temporal bone computed tomography (CT) features are linked to the presence and type of hearing loss in osteogenesis imperfecta (OI) when considering hearing-impaired OI patients and normally hearing (NH) OI ones; (ii) to assess whether other factors influence CT features in a large sample: age, type of mutation, or bone mineral density (BMD).

## 2. Material and Methods

### 2.1. Type of Study

This was a prospective descriptive study conducted at a tertiary referral university hospital.

### 2.2. Subjects

Participants in the present study were selected from among 64 adult OI patients recruited for a clinical trial evaluating the effect of alendronate on BMD [11]. Only 41 accepted to undergo a TB CT scan at the time of the last clinical trial assessment visit. They were 23 males and 18 females, aged 23 to 79 years, with a median of 38 years, and a mean of 40 ± 12 years, at audiometry and CT time. Thus, 82 ears were evaluated. In total, 13 ears (10 patients) had had stapes surgery (Figure 1). A total of 3 patients had a history of hearing loss, which might be explained by other etiologies (head trauma, unilateral TB fracture, and congenital deafness). Mutations of COL1A1 or COL1A2 were identified in 36 patients from the sample.

Inclusion criteria to the clinical trial were reported in Chevrel et al. [11]. OI was defined by the association of at least three criteria:A typical personal history of bone fragility fractures (fractures in the absence of a car crash or major trauma) before 20 years of age, with at least three fragility fractures.At least one criterion in the following list: blue sclerae, scoliosis (the Cobb method with an angle ≥10° but ≤40°), hearing loss (defined by elevated pure-tone thresholds above 20 dB on at least two octave frequencies or a mean air-bone gap above 10 dB), abnormal laxity of ligaments defined by at least three criteria of Carter and Wilkinson, dentinogenesis imperfecta defined by at least a yellow to brown tooth, and at least one family member with OI. [12]A low BMD of the lumbar spine or hip (total femur) measured by dual X-ray densitometry (DXA) with a T score of −2.5 or below at one site.

The exclusion criteria were: evidence of concomitant malignant disease, primary hyperparathyroidism, osteomalacia or other generalized bone diseases or disorders known to influence bone metabolism, renal clearance < 35 mL/mn, pregnancy or lactation, bilateral hip prosthesis, medication using glucocorticoids (≥5 mg of prednisone or equivalent/day), anabolic steroid and/or calcitonin taken during the month prior to the study’s onset, estrogen replacement therapy, and bisphosphonates and fluoride salts during the 6 months before starting the study. None of the patients were exposed to sound trauma before the study.

Ethical considerations: Informed consent was obtained from all participants. The local Ethics Committee (Comité Consultatif de Protection des Personnes dans la Recherche Biomédicale–Lyon B) approved the study.

### 2.3. Methods

#### 2.3.1. Audiometric Evaluation

Hearing tests consisted of pure-tone audiometry by air and bone conduction in standard soundproof conditions. Types of hearing loss were defined to make a possible comparison with the study by Swinnen et al. [10]. A normal pure-tone audiogram was characterized as 0.5 to 4 kHz with air-conduction thresholds below 20 dB. A conductive hearing loss component (CC) was defined by an air-bone gap superior to 10 dB. A sensorineural hearing loss component (SNC) was defined by 0.5 to 4 kHz, air-conduction thresholds above 20 dB, and a 2–4 kHz mean threshold superior to a 0.5–1 kHz mean threshold. These criteria were the same used in the study by Swinnen et al. [10].

#### 2.3.2. Radiologic Evaluation

TB computed tomography was conducted less than 3 months before or after the audiometric evaluation, between 2002 and 2003. All examinations were performed on multislice CT scanners with axial volumetric acquisitions. Axial and coronal reconstructions in lateral semi-circular canal planes were performed at 0.5 to 1 mm thicknesses, with a reconstruction increment of 0.5 mm.

All CT images were blindly read twice by an experienced radiologist (ALB), with a 3-month delay between readings. The patients’ hearing, and clinical and genetic features were not known by the radiologist.

A middle ear assessment included temporal pneumatization, malleus, incus, and stapes density, and footplate (FP) aspect. The last was considered only in non-operated patients because of remodeling after a stapedectomy. Specific attention was given to footplate hypodensity (Figure 2A–E).

Regarding the inner ear, round window obstruction, otic capsule layers (external, middle, and internal layers) and premeatal (i.e., antero-inferior internal meatus) hypodensities, endolabyrinthic extension, and facial nerve canal hypodensity were also carefully evaluated (Figure 2F–O).

#### 2.3.3. Mutations

Mutations of COLIA1 or COLIA2 genes were determined by J. Korkko and D. Prockop from the Tulane Center for Gene Therapy [13]. Stop codon mutations in the COL1A1 gene, mutations resulting in a change in the reading frame of COL1A1, mutations resulting in substitution of glycine by another amino acid in COL1A1 or COL1A2 genes, Gly-Pro-Pro triplet deletion or insertion, variations in an intronic sequence, and mutations in the C-propeptide sequence of COL1A2 were found. The type of mutation suggests putative effects on collagen metabolism [13]. Based on these mutation putative effects, we classified patients in two groups: a group of patients carrying a mutation (stop codon or change in reading frame) believed to result in a purely quantitative decrease in collagen synthesis, and a second group of patients with a mutation expected to induce qualitative changes in collagen fibers (glycine substitution or Gly-Pro-Pro triplet deletion/insertion).

#### 2.3.4. Bone Mineral Density

BMD of the lumbar spine, total femurs (TOT), and femoral necks (Neck) were measured twice (R1 and R2) at baseline of the clinical trial by dual X-ray absorptiometry with a Hologic device (Waltham, MA, USA), as mentioned in the protocol [11]. With this device, the coefficients of variation for each site measured were 0.86% for lumbar spine BMD and 1.04% for total hip BMD.

#### 2.3.5. Statistical Analysis

Statistical analysis was performed using IBM SPSS software (Chicago, IL, USA). Categoric variables were compared between groups with Fisher’s exact X^2^ test. Means were compared using Student’s *t*-test. Statistical significance was defined as a *p* value < 0.05.

## 3. Results

### 3.1. Description of Various CT Features

Labyrinthic and perilabyrinthic aspects were analyzed in 82 ears, but middle ear features were considered in only 69 non-operated ears, as their analysis was not reliable in operated ears.

The first observation was the large heterogeneity of CT scan anomalies, from limited to extended ones (Figure 3, Table 1). Among 69 non-operated ears, 60 (87%) presented both fenestral and retrofenestral anomalies, and 4 ears (6%, from 3 patients) had completely normal CT scans.

Regarding the middle ear, only 69 non-operated ears were considered. Both ossicles and the footplate had a normal aspect in one ear (1%), whereas ossicle hypodensities were observed in 53 ears (77%), FP thickening in 59 ears (23 anterior, 36 extensive; 86%), and footplate hypodensities in 30 ears (43%), all with associated thickening.

This heterogeneity was also seen for labyrinthic and perilabyrinthic bones that were completely normal in 16 out of 82 ears (20%), all non-operated. Middle layer hypodensity was observed in 58 ears (71%); 46 (56%) also had outer layer hypodensity, and 22 (27%) had inner layer hypodensity, thus, three affected layers (Table 1). Premeatal hypodensity was found in 51 ears (62%), oval and round window obstruction in 7 and 15 ears (9 and 18%), respectively, perifacial hypodensity in 7 (9%), and endolabyrinthic extension in 14 ears (17%). All ears with endolabyrinthic extension or round or oval window obstruction showed hypodensities of the outer and middle layer (Table 1).

There was no link between ossicular or fenestral anomalies and retrofenestral abnormality: even if 41 of the 59 ears with fenestral involvement had middle layer hypodensity, 6 ears had a normal footplate, but also had retrofenestral anomalies.

### 3.2. Comparison between CT and Hearing

Hearing features of the sample are presented in Table 2. Comparisons of the TB CT scan with hearing indicated several unexpected results (Figure 4). Footplate anomalies (thickening +/− hypodensity) were found in 49 non-operated ears without a CC (among 59, i.e., 83%; Fischer’s exact bilateral test: *p* = 0.34), as shown in Figure 4A, and in 33 non-operated NH ears (among 40, i.e., 82%). Conversely, there was no ear with a CC and normal footplate.

No perilabyrinthic anomaly on the CT was found to be pathognomonic of an SNC. Conversely, a normal cochlear function can be observed with extensive perilabyrinthic CT anomalies.

When outer or middle layer hypodensity, endolabyrinthic extension, or round and oval windows obturation were observed, there was significantly more often an SNC than when these CT-scan features were absent (Figure 4B: 50% vs. 25%, *p* = 0.03; 47% vs. 21%, *p* = 0.05; 86% vs. 37%, *p* = 0.001; 87% vs. 36%, *p* = 0.0001; and 100% vs. 40%, *p* = 0.003, respectively; bilateral Fisher’s exact tests). All ears with oval window obstruction presented an SNC, and most ears with round window obstruction and/or endolabyrinthic extension had an SNC, except two ears (six patients). The same results were obtained when excluding ears with an SNC possibly due to another etiology (three patients; head trauma, unilateral TB fracture, congenital deafness).

### 3.3. Comparison between CT and Age, Bone Mineral Density and Gene Mutations

The secondary objective of our study was to assess whether age, BMD, and mutation type influence CT features in a large sample of OI patients.

In this sample, there was an association between age and footplate thickening or hypodensity that almost reached statistical significance (median 28 vs. 38 years) (Figure 5A). The age of ears with a normal middle layer CT aspect was significantly lower than that of ears with middle layer hypodensity (35.5 vs. 43 years old).

Regarding BMD, patients with inner layer hypodensity tended to have a lower total right femur BMD than those with a normal inner layer (significant difference of medians, *p* = 0.02, and of distributions, *p* = 0.002; Mann–Whitney test), and those with temporal hyperpneumatization had a lower right femoral neck BMD than those without (difference of medians, *p* = 0.0001, and of distributions, *p* = 0.002; Mann–Whitney test) (Figure 5B). Other BMD variables, especially of the left side, did not differ significantly between groups with or without any TB features.

Finally, regarding mutations, the number of footplate, inner layer, endolabyrinth, and premeatal anomalies was significantly higher in patients with a mutation expected to induce a quantitative decrease in collagen synthesis than in patients with a mutation expected to induce qualitative changes in collagen fibers (94 vs. 72%, *p* = 0.05; 41 vs. 11%, *p* = 0.02; 30 vs. 6%, *p* = 0.04; and 75 vs. 33%, *p* = 0.003, respectively; Fisher’s unilateral Khi2 tests), as shown in Figure 6.

## 4. Discussion

To our knowledge, only one study aimed to compare hearing loss and CT in OI patients [10].

The main results of our study are the high prevalence and the heterogeneity of TB CT anomalies in patients with OI, with or without hearing impairment (HI). Even if some middle ear and otic capsule features were significantly more frequent in ears with hearing loss, they were also observed in many NH ears. No specificity of the stapes footplate (FP) was found in ears with conductive hearing loss, nor of otic capsule anomalies in ears with sensorineural hearing loss.

No significant effect of alendronate on OI hearing was evidenced during the follow-up time of the study [11]. However, the possible effect on the temporal bone in the treated patients was not evaluated. Moreover, in a recent study among women with osteoporosis, the risk of hearing loss was not influenced by bisphosphonate use [14].

Age, BMD, and mutation type seemed to influence CT features, each in a different way. The progression of CT lesions with age is difficult to determine. The middle layers’ hypodensities were more frequent than those of the outer layers, especially in older patients, and there is a tendency towards the effect of age for FP anomalies. Oval window obstruction occurred only in ears with all three layers and premeatal hypodensities. BMD was mainly linked to hyperpneurmatization. The type of collagen mutations, expected to reduce quantity or affect quality of collagen, was found to be linked to perifacial, inner layer, and premeatal density, and to endolabyrinthic extension.

Few publications on the same topic are available to compare our results with. Heterogeneity of TB anomalies and lack of progression from fenestral to retrofenestral features were not reported by Veillon et al. [8], probably because only a few OI cases were observed. The discrepancy with OS also appears in the progression of otic capsule hypodensities, which in our sample started in the middle layer, before the outer layer [15]. Swinnen et al. [10] performed TB CTs according to the patients’ hearing complaints and included only two ears with normal hearing. In our study, patients were referred by rheumatologists for systematic hearing screening and TB CT, with a hearing complaint or not. This might explain why these authors did not observe retrofenestral anomalies in NH ears [10]. However, they suspected discrepancies between fenestral damage and hearing loss with a CC (one NH ear with oval window obstruction), which are largely observed in our study. As in the present sample, Swinnen et al. concluded that TB CT-scan aspects in OI agree less systematically than in OS with the existence and type of HI. In our sample, CT hearing discrepancies existed even when excluding other labyrinthic abnormalities that Swinnen et al. suspected to explain the mixed hearing loss of one of their patients [10].

Our results are the first to introduce significant doubt about correlation between hearing and TB histopathology in OI. Probably because OI is a general bone disease, not limited to the TB, histological involvement is different from OS; therefore, it expresses itself differently on imaging. Otic capsule bone metabolism may be different in OI than in OS; maybe bone remodeling affects inner ear microvascularization or ionic endolymphatic balance, and thus, hearing [16]. OI might also specifically affect the spiral bone next to the organ of Corti and to afferent neurons. Although the exact physiopathology of an SNC in OI is not known, “it is thought to be a consequence of atrophy of the cochlear hair cells and the stria vascularis and abnormal bone remodelling in the cochlea and other labyrinthine structures” [17]. Microfractures in the otic capsule were found in the congenital, and much more severe, form of OI [18]. Hyalinization of the stria vascularis was observed, and primary neural degeneration, potentially secondary to abnormal expression of COL1A2 in membranous cochlear cells, was hypothesized in one histological case [19].

Trying to explain the audio-histological discrepancies, we searched for an effect of age on TB CT; however, that has never been studied in the literature before. This study is the first to point out that TB anomalies may be more frequent depending on the type of collagen mutations. Mutation type is not correlated neither with prevalence, nor age of onset, type, or severity of HI, even with variations within families [20,21,22,23]. Mutation may be a factor in the progression of hearing loss in a pure SNC, but not in a mixed or CC [23]. However, one can imagine that quantitative mutations, expected to decrease collagen formation, induce more severe osteogenesis disturbances; therefore, more otic capsule hypodensities than mutations are expected to alter collagen quality.

Regarding BMD, Swinnen et al. found a lower trabecular BMD in OI patients with conductive or mixed hearing loss than in those with normal hearing or pure sensorineural loss, but comparison between TB CT and BMD has never been reported before our study [24]. In our study, BMD was highly heterogenous between subjects, and BMD differences as a function of TB features were only slight. This heterogeneity might explain why statistical difference is obtained only for the right total BMD and the right femoral neck. However, it seems logical that BMD is higher in ears with no TB anomalies. There are certainly common histological and bone remodeling mechanisms in the mastoid and femoral neck, explaining why femoral neck BMD is lower in ears with temporal hyperpneumatization. One hypothesis should be that the modifications to the auditory ossicles are accompanied by excessive hypermineralization, osteocyte death, and micropetrosis [25]. But it is difficult to understand how inner layer hypodensity could be linked to BMD, as otic capsule ossification is endochondral and finished before birth, contrary to peripheral skeletal ossification.

Technical limits, such as the impossibility to analyze oval windows in operated patients and CT heterogeneous quality, may impact our results. As in Swinnen’s study, CTs were made in different centers and with various devices [10]. However, radiologists used the same CT protocol with fine cuts for all patients of our series. Another pitfall was the imperfect accuracy of audiometry, especially for a mild CC, but this is an inevitable limit of this subjective test. Finally, TB CTs might have been affected by alendronate treatment, which patients took as part of the clinical trial; however, no significant effect on hearing was evidenced during the follow-up time [11,26].

## 5. Conclusions

Our study of TB CT in adult OI patients shows the absence of clear progression from fenestral to retrofenestral lesions, and the absence of extension from the outer to middle layer. Moreover, CT anomalies are frequent even in normally hearing patients and not linked to a type of hearing loss. The imaging differences between OI and OS may be due to their different physiopathology. Indeed, collagen mutations have a global influence on skeletal bone remodeling. Hearing in OI is affected by other factors than just otic capsule remodeling, whereas an anomaly is limited to the TB in OS.

Because progression of TB CT features is highly heterogeneous in OI, CT cannot predict the risk of an SNC, which is, anyway, much higher in OI than in OS. Moreover, CT is not of interest in making a decision about stapes surgery.

## Figures and Tables

**Figure 1 jcm-11-02171-f001:**
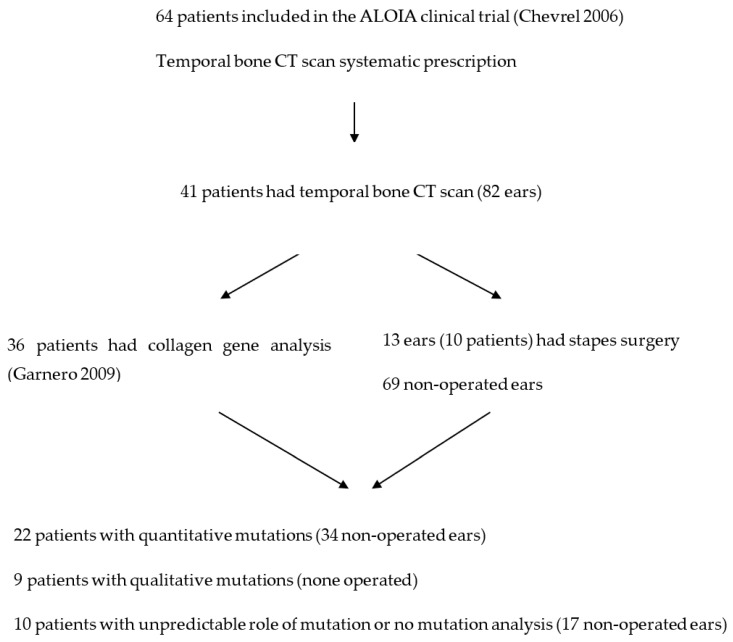
Flow chart.

**Figure 2 jcm-11-02171-f002:**
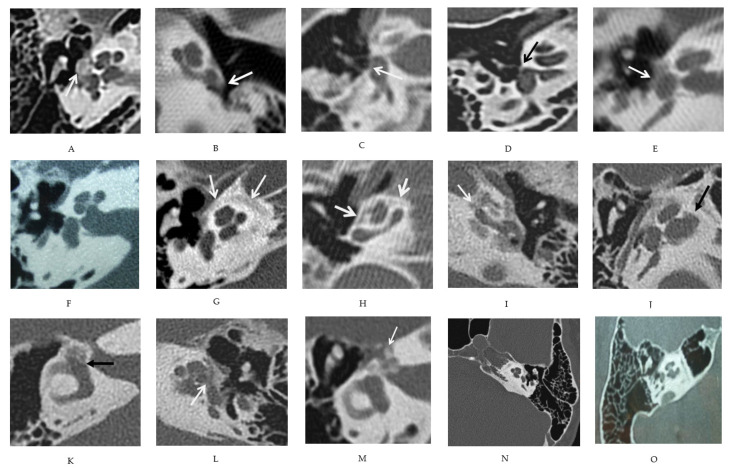
High resolution of temporal bone CT anomalies in OI. Windows: (**A**) oval window obturation by a hypertrophic nodule (white arrow); (**B**) round window obturation (white arrow). Stapes footplate anomalies: (**C**) diffuse hypodense thickening (white arrow); (**D**) anterior thickening (black arrow); (**E**) extensive thickening (white arrow). Otic capsule hypodensities: (**F**) normal otic capsule; (**G**) outer layer hypodensity (white arrow); (**H**) middle layer hypodensity (white arrow); (**I**) inner layer hypodensity, in touch with and extended over the three layers in the cochlea (white arrow). Premeatal hypodensity: (**J**) endolabyrinthic extension; (**K**) endovestibular extension (black arrow); (**L**) endocochlear extension (white arrow). Peri-facial anomalies: (**M**) otic capsule hypodensity around the first portion of facial canal and geniculate ganglion (white arrow). Ossicle hypodensity: (**N**) malleus and incus hypodensity; (**O**) temporal scale and apex petrous hyperpneumatisation.

**Figure 3 jcm-11-02171-f003:**
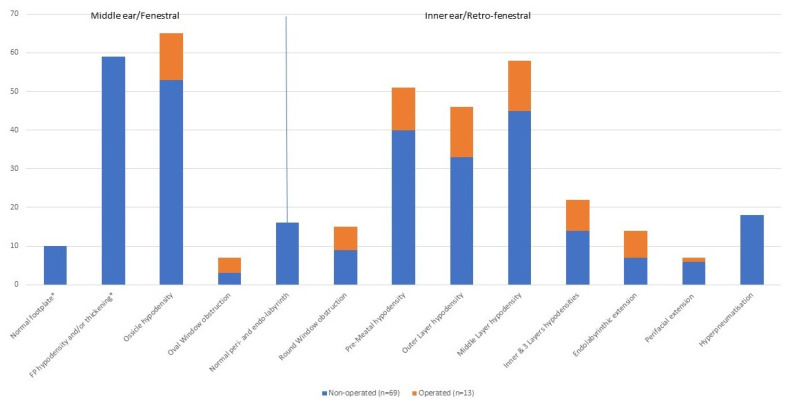
Number of ears with each CT feature *n* = 82 (* stapes footplate (FP) aspect is not presented for operated ears because not interpretable *n* = 69). Hypod. = hypodensity; thick. = thickening; obs. = obstruction.

**Figure 4 jcm-11-02171-f004:**
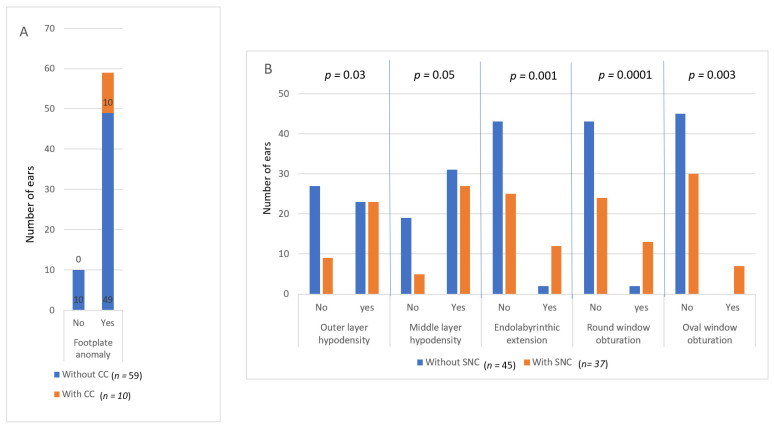
Repartition of CT anomalies compared to hearing. (**A**) Footplate anomalies (thickening and/or hypodensity) in non-operated ears with and without a conductive hearing loss component (CC). (**B**) Peri- and endo-labyrinthic CT anomalies in ears with and without a sensorineural hearing loss component (SNC).

**Figure 5 jcm-11-02171-f005:**
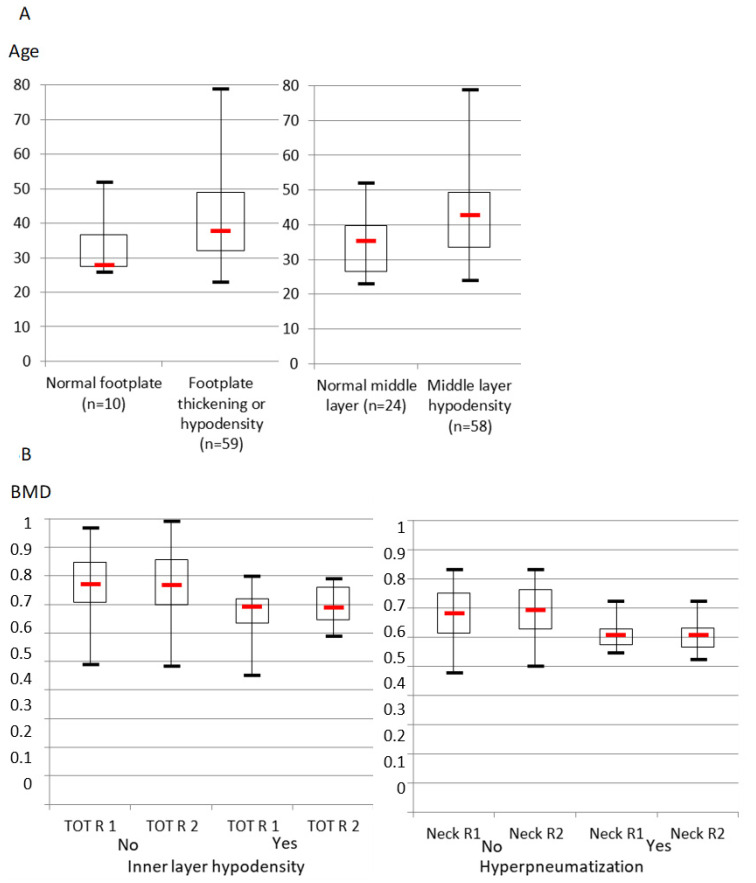
Age and bone mineral density (BMD) values (quartiles, median, extremes) of ears with and without significant CT features. (**A**) Age (years). Left part: with and without footplate thickening or hypodensity (difference of distributions *p* = 0.05; Mann–Whitney test). Right part: with and without middle layer hypodensity (difference of medians *p* = 0.03 and distributions, *p* = 0.004; Mann–Whitney test). (**B**) BMD. Left part: right total BMD without and with inner layer hypodensity (TOT R1 and TOT R2 are two measures of total right femur BMD). Right part: right femoral neck BMD without and with hyperpneumatization (Neck R1 and Neck R2 are two measures of right femoral neck BMD).

**Figure 6 jcm-11-02171-f006:**
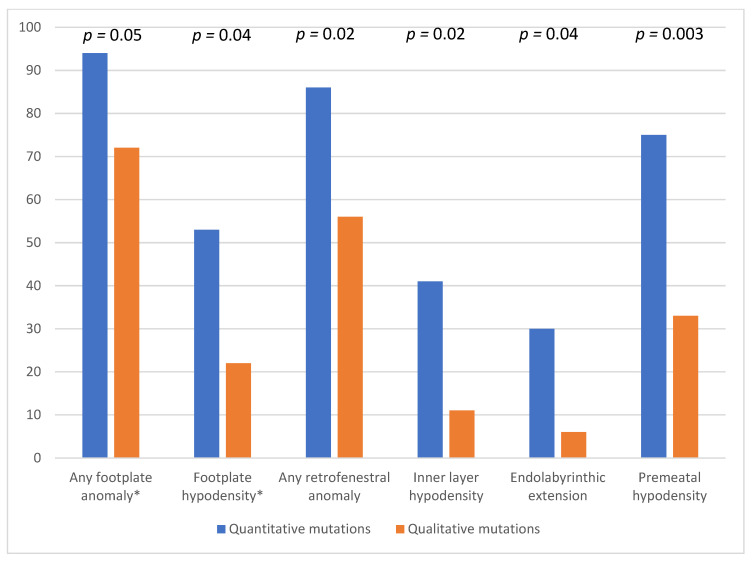
CT features found as being significantly more frequent according to expected effect of COL1 gene mutations. Percentages of ears are relative to number of ears with each type of mutation: 18 ears from subjects with qualitative mutations, 44 ears with quantitative mutations. Stapes footplate aspect is presented only in non-operated ears, i.e., 34 out of 44 ears with quantitative mutations. (* stapes footplate (FP) aspect is not presented for operated ears because not interpretable).

**Table 1 jcm-11-02171-t001:** Number of ears (82 ears) with each CT feature, according to the others. Stapes footplate aspect is not evaluated for operated ears because not interpretable (*n* = 69).

	Normal Footplate	Footplate Hypodensity or Thickening	Ossicle Hypodensity	Oval Window Obstruction	Round Window Obstruction	Pre-Meatal Hypodensity	Outer Layer Hypodensity	Middle Layer Hypodensity	Inner & 3 Layers Hypodensities	Endolabyrinthic Extension	Perifacial Extension	Total
Ossicle hypodensity	9	44										
Oval Window obstruction		3	0									
Normal otic capsule	4	12	14	0								
Round Window obstruction	0	9	13	5								
Pre-Meatal hypodensity	0	36	38	7	15							
Outer Layer hypodensity	2	31	34	7	15	35						
Middle Layer hypodensity	4	41	43	7	15	43	46					
Inner & 3 Layers hypodensities	1	13	18	6	9	17	22	22				22
Endolabyrinthic extension	0	7	11	5	9	12	14	14	14			14
Perifacial extension	0	6	5	2	4	7	6	7	4	2		
Hyperpneumatisation	3	15	14	0	4	14	12	14	0	0	0	18
Total non operated	10	59	53	3								69
Total			65	7	15	51	46	58	22	14	7	82

Bordered and the same grey cells indicate when all ears have the same associated features.

**Table 2 jcm-11-02171-t002:** Number of ears with various hearing features.

	Number of Ears	Normal Hearing	Conductive Component	Sensorineural Component	Both Conductive & Sensorineural Components
Total sample	82	41	11	37	7
Operated	13	1	1	12	1
Non-operated	69	40	10	25	6

## Data Availability

All the data are indicated in the article.

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
