# Peer review of "High Heterogeneity of Temporal Bone CT Aspects in Osteogenesis Imperfecta Is Not Linked to Hearing Loss"

_jcm, 2022, doi:10.3390/jcm11082171_

Round 1

Reviewer 1 Report

Review: “High heterogeneity of temporal bone CT aspects in osteogenesis imperfecta is not linked to hearing loss: Look at CTs with a different eye from otosclerosis!”

Aim of the paper:

  • The objectives are well described and clear

Strengths of the study

  • CT-scan seems not to predict hearing loss in patients with Osteogenesis imperfecta
  • CT-scan pathological findings related with Osteogenesis imperfecta had a significant association with collagen quantitative genes mutation, instead of patients’ age or bone mineral density
  • This is the second study reporting such an experience within the Literature

Major concerns:

  • Study population consisted of patients already recruited for a therapeutic trial, where they were treated with alendronate. Thus, first question is may the alendronate therapy influence the CT findings and hearing function of such patients? This might be a limitation of present study.
  • The authors try to find a relationship between temporal bone CT findings and bone mineral density (BMD) as measured at the femur’s neck. However, a theoretical explanation is still lacking.

Some issues are herein suggested:

  • Please modify the title, which is focusing on the otosclerosis which is not the main topic discussed in the manuscript
  • Please add some other keywords such as Temporal bone computed tomography, Bone mineral density
  • Line 65: “genes” instead of “gene”
  • Line 74-76: “Swinnen et al [10], firstly comparing hearing and CT scan, observed discrepancies between CT and hearing in 27.8% (17 patients).”
  • Line 97: It is preferable to explain here the inclusion criteria
  • Line 98: “Local” instead of “local”
  • Line 128-129: “to make a possible comparison with” instead of “to make comparison possible”
  • Line 130-133: Explain better the definitions of the different types of hearing loss
  • Line 144: “footplate (FP)” instead of “FP”
  • Line 145-147: The footplate thickening classification herein proposed is no more mentioned during the following entire article. Please explain its usefulness for this study. Is it a validated classification?
  • Line 169: “mutations resulting” instead of “mutation resulting”
  • Line 176: “and a second group of patients” instead of “and a second group the patients”
  • Line 182: explain better what it means the word “CV” and add it to the abbreviation’s list
  • Line 253-254: “In this sample, there was an association between age and footplate thickening or density that almost reached statistical significance”.
  • Please specify in material and methods subheading the measures of the BMD that are reported in the results subheading “TOT R1, TOT R2, Neck R1 and Neck R2”
  • Line 273: explain better what the authors mean with “proportion of footplate, etc.”
  • Line 296-298: it is better to simplify this sentence; we can suggest this way: “Middle layers hypodensities are more frequent than those of outer layers, especially in older patients, and there is a tendency towards the effect of age for FP anomalies”.
  • Line 302: “to be linked to” instead of “to be liked to”
  • Line 309-312: It is better to explain this concept with a different organization of the sentence in order to make clear that Swinnen et al. performed TB CTs according to the patient’s hearing complain
  • Line 335: “Trying to explain” instead of “To try to explain”
  • Line 371-374: Indeed… this sentence is not clear please rephrase
  • Line 375-377: it is better to split the first part of the sentence from the second one. For example: “CT cannot predict the risk of SNC which is, anyway, much higher in OI than in OS. Moreover, CT isn’t of interest in making decision about stapes surgery”

Author Response

Thank very much for the comments of the reviewer 1. The answers are indicated in bold in the new version. We hope that these corrections will satisfy the reviewer and the redaction of the Journal of Clinical Medicine.

Major concerns:

Study population consisted of patients already recruited for a therapeutic trial, where they were treated with alendronate. Thus, first question is may the alendronate therapy influence the CT findings and hearing function of such patients? This might be a limitation of present study.

The influence of alendronate therapy on the CT findings and hearing function in OI patients is still unknow. In recent study among women with osteoporosis, the risk of hearing loss was not influenced by bisphosphonate use. This paragraph has been added in the discussion and a reference has been added:

  No significant effect of alendronate on OI hearing has been evidenced during the follow-up time of the study [11]. However, the possible effect on the temporal bone in the treated patients has been not evaluated. Moreover, in a recent study among women with osteoporosis, the risk of hearing loss was not influenced by bisphosphonate use [14].

The authors try to find a relationship between temporal bone CT findings and bone mineral density (BMD) as measured at the femur’s neck. However, a theoretical explanation is still lacking.

BMD of the lumbar spine, total femurs and femoral necks were measured at baseline of the clinical trial. As mentioned in results, patients with inner layer hypodensity tended to have a lower total right femur BMD than those with normal inner layer Other BMD variables, especially of the left side, did not differ significantly between groups with or without any TB features.  Few studies evaluating the structure of ossicles from OI patients have been published. One hypothesis should be the modifications auditory ossicles, accompanied by excessive hypermineralization, osteocyte death and micropetrosis. This point has been discussed and reference added in discussion.

Some issues are herein suggested:

Please modify the title, which is focusing on the otosclerosis which is not the main topic discussed in the manuscript.

Title has been modified as suggested by the Reviewer. The title has been simplified and the focus on otosclerosis suppressed: High heterogeneity of temporal bone CT aspects in osteogenesis imperfecta is not linked to hearing loss.

Please add some other keywords such as Temporal bone computed tomography, Bone mineral density.

These keywords have been added.

Line 65: “genes” instead of “gene”

This sentence has been corrected.

Line 74-76: “Swinnen et al [10], firstly comparing hearing and CT scan, observed discrepancies between CT and hearing in 27.8% (17 patients).”

This sentence has been modified as suggested by the reviewer.

Line 97: It is preferable to explain here the inclusion criteria

Paragraph has been modified as suggested by reviewer.

Line 98: “Local” instead of “local”

This sentence has been corrected.

Line 128-129: “to make a possible comparison with” instead of “to make comparison possible”

This sentence has been modified.

Line 130-133: Explain better the definitions of the different types of hearing loss

As already mentioned in our text, we used the same criteria of hearing loss than in the reference 10 by Swinnen to allow the comparison of our studies.  This point has been clarified: These criteria were the same using in the study by Swinnen et al.

Line 144: “footplate (FP)” instead of “FP”

This sentence has been modified.

Line 145-147: The footplate thickening classification herein proposed is no more mentioned during the following entire article. Please explain its usefulness for this study. Is it a validated classification?

As indicated by the reviewer, this sentence is not appropriate because not more used and mentioned during the following entire article.

This sentence:”Footplate thickening was scaled in three grades: anterior, total, or osteodystrophic block by a hypertrophic nodule that blocks up oval win-dow. A specific attention was given to footplate hypodensity (figure 2A-E).” has been changed: “A specific attention was given to footplate thickening and hypodensity (figure 2A-E).

Line 169: “mutations resulting” instead of “mutation resulting”

This sentence has been corrected.

Line 176: “and a second group of patients” instead of “and a second group the patients”

This sentence has been corrected.

Line 182: explain better what it means the word “CV” and add it to the abbreviation’s list

This sentence has been modified as suggested by the reviewer: coefficients of variation for each site measured

Line 253-254: “In this sample, there was an association between age and footplate thickening or density that almost reached statistical significance”.

This sentence has been modified as suggested by the reviewer.

Please specify in material and methods subheading the measures of the BMD that are reported in the results subheading “TOT R1, TOT R2, Neck R1 and Neck R2”

As suggested by the reviewer, the meaning of TOT and Neck have been précised in material and methods: BMD of the lumbar spine, total femurs (TOT) and femoral necks (Neck) were measured twice (R1 and R2) at baseline of the clinical trial by dual X-ray absorptiometry with an Hologic device (Waltham, MA, USA) as mentioned in the protocol

Line 273: explain better what the authors mean with “proportion of footplate, etc.”

The term proportion is mis chosen. It has been modified by number.

Line 296-298: it is better to simplify this sentence; we can suggest this way: “Middle layers hypodensities are more frequent than those of outer layers, especially in older patients, and there is a tendency towards the effect of age for FP anomalies”.

This sentence has been modified as suggested by the reviewer.

Line 302: “to be linked to” instead of “to be liked to”

This sentence has been corrected.

Line 309-312: It is better to explain this concept with a different organization of the sentence in order to make clear that Swinnen et al. performed TB CTs according to the patient’s hearing complain

Line 309-312 has been modified to better explain the two organizations: Swinnen et al. [10] performed TB CTs according to the patient’s hearing complain including only 2 ears with normal hearing. In our study, patients were referred by rheumatologists for systematic hearing screening and TB CT, with hearing complaint or not.

Line 335: “Trying to explain” instead of “To try to explain”

This sentence has been modified

Line 371-374: Indeed… this sentence is not clear please rephrase

This sentence has been modified as suggested by the reviewer: Our study of TB CT in adult OI patients shows the absence of clear progression from fenestral to retrofenestral lesions, the absence of extension from outer to middle layer. Moreover, CT anomalies are frequent even in normally hearing patients and not linked to the type of hearing loss. The imaging differences between OI and OS may be due to their different physiopathology. Indeed, collagen mutations having a global influence on skeletal bone remodeling. Hearing in OI is affected by other factors than otic capsule remodeling whereas anomaly is limited to TB in OS.

Line 375-377: it is better to split the first part of the sentence from the second one. For example: “CT cannot predict the risk of SNC which is, anyway, much higher in OI than in OS. Moreover, CT isn’t of interest in making decision about stapes surgery”

This sentence has been modified as suggested by the reviewier: Because progression of TB CT features is highly heterogeneous in OI, CT cannot predict the risk of SNC which is, anyway, much higher in OI than in OS. Moreover, CT is not of interest in making decision about stapes surgery.

Reviewer 2 Report

Many thanks for the opportunity to review this paper. I found this study very interesting, well documented, and comprehensive. The majority of hearing loss cases described in the sample had a sensorineural component (37 out of 82 ears); I’m wondering how many of these cases can be attributed to ageing, since the sample was aged in a range of 23-79. Probably the median age with the relative standard deviation could be added. Noise exposure should also be considered among exclusion criteria. Furthermore, I deduce 36 patients were evaluated for genetic mutations and 31 among these showed quantitative/qualitative mutations. At line 95, I only suggest modifying the term “identified” with “investigated” or similar, because the sentence can be misinterpreted.

Author Response

Thank very much for the comments of the reviewer 2. The answers are indicated in bold in the new version. We hope that these corrections will satisfy the reviewer and the redaction of the Journal of Clinical Medicine.

I’m wondering how many of these cases can be attributed to ageing, since the sample was aged in a range of 23-79. Probably the median age with the relative standard deviation could be added.

The median age with relative standard deviation has been added: median 38 years, mean 40 ± 12 years.

Noise exposure should also be considered among exclusion criteria.

Any patient include in our study has been exposed to noise. This exclusion criteria has been added as suggested by reviewer 2: None of patients was exposed to sound trauma before the study.

Furthermore, I deduce 36 patients were evaluated for genetic mutations and 31 among these showed quantitative/qualitative mutations. At line 95, I only suggest modifying the term “identified” with “investigated” or similar, because the sentence can be misinterpreted.

This sentence has been modified as suggested by the reviewer 2.

Round 2

Reviewer 1 Report

Great job